# Curcumin Suppresses TGF-β1-Induced Myofibroblast Differentiation and Attenuates Angiogenic Activity of Orbital Fibroblasts

**DOI:** 10.3390/ijms22136829

**Published:** 2021-06-25

**Authors:** Wei-Kuang Yu, Wei-Lun Hwang, Yi-Chuan Wang, Chieh-Chih Tsai, Yau-Huei Wei

**Affiliations:** 1Institute of Clinical Medicine, National Yang Ming Chiao Tung University, Taipei 112, Taiwan; wkyu3@vghtpe.gov.tw; 2Department of Ophthalmology, Taipei Veterans General Hospital, Taipei 112, Taiwan; 3Department of Biotechnology and Laboratory Science in Medicine, National Yang Ming Chiao Tung University, Taipei 112, Taiwan; a85296658@gmail.com (W.-L.H.); bill0513a@gmail.com (Y.-C.W.); 4Cancer Progression Research Center, National Yang Ming Chiao Tung University, Taipei 112, Taiwan; 5Program in Molecular Medicine, National Yang Ming Chiao Tung University, Taipei 112, Taiwan; 6Center for Mitochondrial Medicine and Free Radical Research, Changhua Christian Hospital, Changhua City 500, Taiwan

**Keywords:** curcumin, Graves’ ophthalmopathy, orbital fibrosis, thyroid eye disease

## Abstract

Orbital fibrosis, a hallmark of tissue remodeling in Graves’ ophthalmopathy (GO), is a chronic, progressive orbitopathy with few effective treatments. Orbital fibroblasts are effector cells, and transforming growth factor β1 (TGF-β1) acts as a critical inducer to promote myofibroblast differentiation and subsequent tissue fibrosis. Curcumin is a natural compound with anti-fibrotic activity. This study aims to investigate the effects of curcumin on TGF-β1-induced myofibroblast differentiation and on the pro-angiogenic activities of orbital fibroblasts. Orbital fibroblasts from one healthy donor and three patients with GO were collected for primary cell culture and subjected to myofibroblast differentiation under the administration of 1 or 5 ng/mL TGF-β1 for 24 h. The effects of curcumin on TGF-β1-induced orbital fibroblasts were assessed by measuring the cellular viability and detecting the expression of myofibroblast differentiation markers, including connective tissue growth factor (CTGF) and α-smooth muscle actin (α-SMA). The pro-angiogenic potential of curcumin-treated orbital fibroblasts was evaluated by examining the transwell migration and tube-forming capacities of fibroblast-conditioned EA.hy926 and HMEC-1 endothelial cells. Treatment of orbital fibroblasts with curcumin inhibited the TGF-β1 signaling pathway and attenuated the expression of CTGF and α-SMA induced by TGF-β1. Curcumin, at the concentration of 5 μg/mL, suppressed 5 ng/mL TGF-β1-induced pro-angiogenic activities of orbital fibroblast-conditioned EA hy926 and HMEC-1 endothelial cells. Our findings suggest that curcumin reduces the TGF-β1-induced myofibroblast differentiation and pro-angiogenic activity in orbital fibroblasts. The results support the potential application of curcumin for the treatment of GO.

## 1. Introduction

Orbital fibrosis, a hallmark of tissue remodeling in Graves’ ophthalmopathy (GO), is an appearance-destructing and vision-threatening condition that is mostly associated with Graves’ disease (GD). Although the mechanism underlying the thyroid endocrine dysfunction in GD is relatively better elucidated, the pathophysiology of GO is incompletely understood [1]. The pathophysiology of GO is thought to be a complex interaction between autoimmune response and reactive oxygen species (ROS) [2]. The clinical manifestations of GO are associated with inflammatory enlargement of orbital soft tissues characterized by mononuclear cell infiltration, adipogenesis, and orbital fibrosis [3,4]. Orbital fibrosis in advanced GO may result in severe morbidities, including exophthalmos, exposure keratitis, diplopia, and dysthyroid optic neuropathy, which are often unresponsive to drug treatment and require surgical intervention [5]. Investigating the molecular mechanisms modulating the myofibroblast differentiation in GO is pivotal to the development of novel therapeutic strategies for blocking orbital fibrosis.

The roles of transforming growth factor β1 (TGF-β1) in myofibroblast activation and extracellular matrix production have been widely known to initiate tissue fibrosis in various types of cells and organs [6,7,8,9]. Previous studies showed increased expression of TGF-β1 in surgically obtained orbital soft tissue [10] and primary culture of orbital fibroblasts [11] of GO compared to those of normal controls, suggesting a clinicopathologic role of TGF-β1 in the disease progression of orbital fibrosis. In addition, TGF-β1 can trigger the differentiation of CD90-positive orbital fibroblasts into myofibroblasts that express α-smooth muscle actin (α-SMA) [12]. However, the role of TGF-β1 in the mechanism of orbital fibrosis has remained poorly understood.

Angiogenic activities are involved in the pathogenesis associated with myofibroblast activation and differentiation, including wound healing, diabetic retinopathy, chronic inflammatory diseases, and cancer [13,14]. It is not clear whether angiogenic activities of orbital fibroblasts affect the fibrotic process of the GO disease.

Curcumin is a well-known natural compound extracted from the root of *Curcuma longa*. Numerous studies have proven its wide range pharmacological effects, including anti-inflammation, anti-oxidation, fibrosis reduction, and angiogenesis modulation [15,16,17]. In addition, curcumin as an oral supplement has already been used in various medical conditions [18,19,20]. This study aims to examine the curcumin-mediated effect on TGF-β1-stimulated signaling and growth of orbital fibroblasts.

## 2. Results

### 2.1. Establishment of an In Vitro Fibrotic Cell Model of Orbital Fibroblasts

Previous studies have demonstrated the role of TGF-β1 as an important mediator in fibrosis induced by oxidative stress in orbital fibroblasts from patients with GO [11,21]. In an attempt to establish a fibrosis model to identify the anti-fibrotic effects of natural compounds, we isolated and cultured orbital fibroblasts from two donors (donor number one: a healthy subject; donor number two: a patient with GO) in the presence of transforming growth factor β1 (TGF-β1) as the TGF-β1/connective tissue growth factor (CTGF) axis contributes to tissue fibrosis [22]. We found that the administration of 1 ng/mL or 5 ng/mL of TGF-β1 did not alter the viability of fibroblasts from both donors in either 10% FBS-containing growth medium (Figure 1A), or under serum starvation condition (Figure 1B). The expression of CTGF and α-SMA increased in a dose-dependent manner upon TGF-β1 treatment (Figure 1C, D) in two culture conditions, suggesting a successful induction of the myofibroblast differentiation in the primary culture of orbital fibroblasts. Owing to the higher expression levels of CTGF and α-SMA when cells were treated with 5 ng/mL TGF-β1 under 10% FBS-containing growth medium culture, we evaluated the anti-fibrotic effect of selected natural compounds under this experimental condition.

### 2.2. Curcumin Suppresses the TGF-β1 Signaling and Inhibits the TGF-β1-Induced Myofibroblast Differentiation of Orbital Fibroblasts

The reactive oxygen species (ROS) are byproducts of mitochondrial respiration in cell metabolism. The imbalanced ROS production elicits apoptosis and mitochondrial dysfunction [23]. ROS production, and the concomitant oxidative stress, also contribute to the activation of various cytokines, including TGF-β1 [24]. Curcumin is a phytopolyphenol mainly found in *Curcuma longa*, which could modulate the intracellular ROS level and autophagy [25]. We first investigated the impact of curcumin on the myofibroblast differentiation of expanded orbital fibroblasts. We found that curcumin administration alone did not change the viability of orbital fibroblasts from two donors (Figure 2A). However, co-treatment of orbital fibroblasts with 5 ng/mL of TGF-β1 and a higher dose of curcumin (5 μg/mL) for 24 h slightly decreased cell viability (Figure 2A). It was found that curcumin treatment suppressed the TGF-β1-induced expression of CTGF and α-SMA in a dose-dependent manner (Figure 2B,C). Moreover, pretreatment with curcumin (5 μg/mL) overnight reduced the amount of phosphorylation of Smad2/3 upon TGF-β1 treatment for 60 min in orbital fibroblasts from Donor-1 (Figure 3A) and Donor-2 (Figure 3B). The attenuating effect of curcumin on TGF-β1-induced myofibroblast differentiation and signaling cascade was also confirmed in GO Donor-3 and -4 (Appendix A). This result suggests that curcumin can inhibit the TGF-β1-signaling cascade of myofibroblast differentiation of GO orbital fibroblasts.

### 2.3. Curcumin Attenuates the TGF-β1-Elicited Migration and Tube-Forming Capacities of EA.hy926 Endothelial Cells

As TGF-β1 is indispensable for vascular development [26] and myofibroblasts contribute to the vascular network development during wound healing [14], we next examined the effects of curcumin on the angiogenic activity of TGF-β1-differentiated orbital fibroblasts. The conditioned media from two orbital fibroblasts administrated with TGF-β1, curcumin alone or co-treatment with TGF-β1 and curcumin were harvested, respectively, to treat the EA.hy926 endothelial cells (ECs). The EA.hy926 cell line demonstrated highly differentiated functions that are characteristic of human vascular endothelium, while offering the advantage of immortality, stability through subculture and the reproducibility of results [27,28]. After 48 h of treatment with the fibroblast-conditioned medium, the viability of ECs was not altered (Figure 4A). To evaluate the pro-angiogenic activity of TGF-β1-treated orbital fibroblasts, the conditioned medium-treated EA.hy926 ECs were subject to the transwell tube-forming assay and migration assay. We found that the conditioned medium derived from curcumin-treated orbital fibroblasts suppressed the TGF-β1-enhanced tube-forming capacity (Figure 4B and Appendix A) and transwell migration capacity (Appendix A) of EA.hy926 ECs, indicating the anti-angiogenic effect of curcumin on orbital fibroblasts treated with TGF-β1. The conditioned medium from curcumin-treated orbital fibroblasts also attenuated the TGF-β1-induced migratory ability and tube-forming capacity in a dermal microvascular endothelial cell line, HMEC-1 (Appendix A).

## 3. Discussion

The activation of orbital fibroblasts, followed by myofibroblast differentiation and excess production of extracellular matrix proteins, are hallmarks of fibrotic disorders [29]. Orbital fibroblasts are the key effector cells in the development of orbital fibrosis and have been found to be widely involved throughout the pathogenesis of GO [30]. TGF-β1 has been reported to promote myofibroblast differentiation and extracellular matrix production in various ocular cell types, including corneal fibroblasts [31], conjunctival fibroblasts [32], and retinal pigmented epithelial cells [33]. On the other hand, CTGF was found to modulate myofibroblast activation, extracellular matrix deposition, angiogenesis, and tissue remodeling [34]. Previous studies have demonstrated a higher baseline CTGF expression level in GO orbital fibroblasts compared to the normal control [35], and the expression level of CTGF of GO orbital fibroblasts was found to be significantly associated with the clinical progression of GO [36]. In addition, the essential role of CTGF in TGF-β1-induced myofibroblast differentiation of GO orbital fibroblasts has been confirmed by knockdown of CTGF via small hairpin RNA of the CTGF gene [22]. Thus, TGF-β1-induced expression of CTGF and α-SMA could be viewed as myofibroblast differentiation biomarkers of orbital fibrosis in GO. Our results confirmed a higher TGF-β1-induced myofibroblast differentiation of GO orbital fibroblasts compared to the normal control, which is a feasible platform to screen for anti-fibrotic compounds (Figure 1). A strong upper band near 40 kDa was detected using the Abcam CTGF antibody (ab6992, Abcam Inc., Cambridge, UK) in Figure 1C,D Exp 1. In a previous study [37], an upper band near CTGF (~38 kDa) in Western blots using Abcam CTGF antibody (ab6992, Abcam Inc., Cambridge, UK) had also been detected in fibroblasts treated with TGF-β, and the expression of these upper bands did not correlate with the expression of CTGF. Although the definite character of this upper band is not clear, we considered it a nonspecific band. We used one additional CTGF antibody (sc-365970, Santa Cruz Inc., (Dallas, TX, USA) to validate the expression of CTGF and, as expected, the administration of TGF-β1 promoted the expression of CTGF (Figure 1C,D Exp 2).

Curcumin has been widely documented as a potential drug for the treatment of tissue fibrosis. Previous studies demonstrated that the fibrosis-blocking mechanism in curcumin is related to its effects of anti-inflammation [38], antioxidation [39], and reduction of extracellular matrix production [40]. In cell and animal studies, curcumin has the ability to block multiple sites of the TGF-β signaling in renal cells [41], suppressing bleomycin-induced pulmonary fibrosis in rats [42], and inhibiting TGF-β1-induced CTGF expression in human gingival fibroblasts [9]. To the best of our knowledge, the curcumin-mediated effect on orbital fibrosis has not been investigated. In this study, we demonstrated that curcumin can suppress the TGF-β1-induced CTGF expression and myofibroblast differentiation in both GO orbital fibroblasts and the normal control with very low cytotoxicity (Figure 2). With the fact that a higher TGF-β1-induced myofibroblast differentiation of GO orbital fibroblasts compared to the normal control (Figure 1), our results suggest the potential of the application of curcumin as an anti-fibrotic agent in the treatment of GO.

We also noted that pretreatment with curcumin (5 μg/mL) overnight reduced the amount of phosphorylation of Smad2/3 upon TGF-β1 treatment (Figure 3), suggesting that curcumin can inhibit the TGF-β1-signaling cascade in orbital fibroblasts. However, there are differences in the Smad2/3 protein expression levels between Donor-1 (Figure 3A) and Donor-2 (Figure 3B). We considered it might be due to the cell-to-cell difference, and further study is needed to figure it out.

The angiogenic activities of orbital fibrosis are poorly understood. Recent studies in pulmonary fibrosis [43], liver fibrosis [44], and cardiac fibrosis [45] revealed an important role of angiogenic activity associated with fibrotic progression. The observation that the clinical activity score is associated with an elevation of serum vascular endothelial growth factor (VEGF) in patients with GO implicated that angiogenesis plays a role in the pathophysiology of orbital fibrosis [46]. We found in this study that a conditioned medium derived from curcumin-treated orbital fibroblasts suppressed the TGF-β1-enhanced transwell migratory ability and tube-forming capacity in EA.hy926 ECs (Figure 4) and in HMEC-1 cells (Appendix A), indicating the anti-angiogenic effect of curcumin on orbital fibroblasts treated with 5 ng/mL TGF-β1. However, the molecular mechanism of action of TGF-β1 in exerting pro-angiogenic activities on myofibroblast differentiation warrants further investigation.

The major limitation of this study is the availability of orbital fibroblasts from only one healthy subject and three GO patients, meaning that we are not able to tell the differences of the curcumin effect on TGF-β1-induced myofibroblast differentiation and angiogenic activities between GO and normal orbital fibroblasts. Nevertheless, we demonstrated the reproducible in vitro effect of curcumin on attenuating TGF-β1-induced myofibroblast differentiation and the angiogenic effect on primary cultures of human orbital fibroblasts. Further experiments, by using orbital fibroblasts from more GO patients, are needed to strengthen our viewpoints and provide further support to the conclusion of this study.

Despite the fact that curcumin has promising potential in the treatment of GO, its clinical applications have been limited by its poor bioavailability in terms of poor solubility and poor absorption in the free form in the gastrointestinal tract. Curcumin is an oil-soluble biologically active compound in concentrates of turmeric, and practically insoluble at room temperature in water at an acidic or neutral pH. Recently, various approaches have been developed in curcumin formulation to enhance its absorption, including consuming curcumin with lipid additions (BCM-95^®^) [47], dispersion of curcumin on matrices (Cavacurmin^®^; Meriva^®^) [48,49], and particle size reduction of curcumin (BioCurc^®^; Theracurmin^®^) [50,51]. Among them, nanoparticle formulations of curcumin showed greatly enhanced absorption resulting in desirable blood levels [52], making it possible to address the therapeutic potential of curcumin in the treatment of GO.

## 4. Materials and Methods

### 4.1. Reagents

Curcumin, also known as diferuloylmethane (1E,6E)-1,7-bis(4-hydroxy-3-methoxyphenyl) hepta-1,6-diene-3,5-dione), (#C7727), anti-GAPDH antibody (#G5262), and anti-β-actin antibody (#A1978), were purchased from Sigma–Aldrich Chemical Co. (St. Louis, MO, USA). Stock solutions of curcumin were prepared in dimethyl sulfoxide (DMSO). Recombinant human transforming growth factor-beta 1 (TGF-β1) (#PO1137) was purchased from R&D Systems, Inc. (Minneapolis, MN, USA). Antibodies specific to CTGF (#ab6992), fibronectin (#ab2413), and α-SMA (#ab5694) were purchased from Abcam Inc. (Cambridge, UK). The additional antibody to CTGF (sc-365970) was purchased from Santa Cruz Inc. (Dallas, TX, USA). Primary antibodies against phospho-Smad2 (Ser465/467)/Smad3 (S423/425) (#8828S) and Smad2/3 (#8685S) were purchased from Cell Signaling Technology (Danvers, MA, USA).

### 4.2. Tissue Dissociation and Cell Culture

The primary cultures of orbital fibroblasts were established from a normal subject who received orbital surgery for noninflammatory conditions (Donor-1) and three GO patients (Donor-2, -3, and -4) receiving orbital decompression surgery in accordance with the Declaration of Helsinki, and with informed consent of the donors (IRB VGHTPE: 2019-08-012B). The GO patients fulfilled the following three requirements: (1) adult (aged 20 years or older) (2) stable euthyroidism for at least 6 months, and (3) maintained at the inactive stage of GO without systemic use of steroids or radiotherapy for a least 1 month before surgery. Exclusion criteria were any ocular diseases other than GO, pregnancy, or chronic diseases, such as diabetes mellitus, diseases of the lung, liver or kidney, cancer, other endocrine dysfunction, and inflammatory disorders. The obtained orbital tissues were cut aseptically into small pieces in phosphate-buffered saline (PBS, pH 7.3), and incubated with a sterile solution containing 130 U/mL of collagenase and 1 mg/mL of dispase (Sigma–Aldrich Chemical Co., St. Louis, MO, USA) for 24 h at 37 °C in an incubator with an atmosphere of 5% CO_2_. The mixture of digested orbital tissues was pelleted by centrifugation at 1000× *g* for 5 min and then resuspended in Dulbecco’s Modified Eagle’s Medium (DMEM, Gibco Life Technologies, Gaithersburg, MD, USA) containing 10% fetal bovine serum (FBS) and antibiotics (100 U/mL of Penicillin G and 100 μg/mL of Streptomycin sulfate, Biological Industries, Kibbutz Beit Haemek, Israel). The cultured orbital fibroblasts were used between the 3rd and 6th passages, and the cell cultures at the same passage numbers were used for performing the same set of experiments. The EA.hy926 endothelial cells and dermal microvascular endothelial cell (HMEC-1) were initially obtained from ATCC (Manassas, VA, USA) and cultured in DMEM supplemented with 10% FBS, 1% NEAA, GlutaMAX, and 1% sodium pyruvate (all from Gibco Life Technologies, Gaithersburg, MD, USA) without antibiotics. The cell culture plates for HMEC-1 were coated with 1 mg/mL of fibronectin (1:100, Sigma–Aldrich Chemical Co., St. Louis, MO, USA) at 37 °C for at least 30 min. The fresh basal DMEM was added to TGF-β1- or curcumin-treated orbital fibroblasts for 24 h to collect the conditioned medium. Approximately 2 × 10^5^ EA.hy926 cells were seeded in wells of a 6-well plate overnight and cultured under a diluted conditioned medium (CM: complete DMEM = 1.5:1) for two days prior to the functional assay. Approximately 3 × 10^5^ HMEC-1A cells were seeded in wells of a 6-well plate overnight and cultured under a diluted conditioned medium (CM: complete DMEM = 2:1) for 24 h prior to the functional assay. All cells were cultured at 37 °C in a humidified incubator with an atmosphere of 5% CO_2_.

### 4.3. Cell Viability

Cells were trypsinized, and the cell number was calculated by Trypan blue exclusion to assess the relative viability.

### 4.4. Western Blot

Total cell proteins were harvested on ice with the RIPA lysis buffer, protease inhibitors, and phosphatase inhibitors (Thermo Fisher Scientific, Waltham, MA, USA). The recipe of the RIPA lysis buffer: 1.5 mL NaCl (1M), 0.1 mL Nonidet P-40 (1%), 0.05 mL sodium deoxycholate (0.5%), 0.01 mL SDS (0.1%), 5 mL Tris-HCl (50 mM, pH 7.4), diluted with ddH_2_O to 10 mL. This solution was stored at 4 °C. The supernatants were collected after centrifugation at 10,000× *g* at 4 °C for 15 min, and the protein concentration was determined using the Bradford Protein Assay Kit (Thermo Fisher Scientific, Waltham, MA, USA). Protein extract was separated by SDS-PAGE and blotted onto a piece of PVDF membrane (Amersham Pharmacia Biotech Inc., Buckinghamshire, UK). After blocking with 5% skim milk in the TBST buffer (50 mM Tris-HCl, 150 mM NaCl, 0.1% Tween 20, pH 7.4) at room temperature for 1 h, the membrane was incubated with the primary antibody at 4 °C overnight. The membrane was then incubated with a horseradish peroxidase (HRP)-conjugated secondary antibody for another 1 h at room temperature, after washing three times with the TBST buffer. The protein signals were visualized by enhanced chemiluminescence (ECL) and detected in an ImageQuant LAS 4000 chemiluminescence detection system (GE Healthcare Bio-Sciences, Piscataway, NJ, USA) or ImageScanner III with the LabScan 6.0 software (GE Healthcare Bio-Sciences, Piscataway, NJ, USA). All uncropped blots are provided in Appendix A. Relative intensities of protein bands were quantified by scanning densitometry, using the Image J software.

### 4.5. Transwell Migration Assay and Tube Formation Assay

For the transwell cell migration assay, 2 × 10^4^ EA.hy926 or 3 × 10^4^ HMEC-1 cells were suspended in 100 µL (EA.hy926) or 200 µL (HMEC-1) of the basal DMEM introduced into the upper chamber of a transwell device with a membrane of 8.0 µm pores (Corning, 3422, NY, USA). An aliquot of 600 µL of 20% FBS-containing medium was added to the lower chamber to create a serum gradient. After 24 h, the migrated cells were fixed by 4% paraformaldehyde (Sigma–Aldrich Chemical Co., St. Louis, MO, USA), and were then stained with 0.1% crystal violet (Sigma–Aldrich Chemical Co., St. Louis, MO, USA) for quantification in 10× field by an inverted light microscope (Nikon, Tokyo, Japan). For the tube formation assay, 20 µL of growth factor-reduced Matrigel (BD, Franklin Lakes, NJ, USA) was spread into the wells of a 96-well plate at 37 °C for 30 min for solidification. Approximately 1 × 10^4^ EA.hy926 cells or 3 × 10^4^ HMEC-1 cells were suspended in 100 µL of complete DMEM and seeded onto a reconstituted Matrigel basement membrane for 6 h (EA.hy926) or 24 h (HMEC-1) at 37 °C. The vascular structures formed were quantified by visual inspection using an inverted light microscope in a 4× field (Nikon, Tokyo, Japan).

### 4.6. Statistics

The SPSS for Mac, version 24.0 (SPSS, Inc., Chicago, IL, USA) was used to analyze the results (Appendix A). The normality of the data was checked by the Shapiro–Wilk test, and all data were distributed in a normal way (*p* > 0.05). The significance level of the difference between groups involved was determined by the ANOVA or independent *t*-test, with the Bonferroni test as a post hoc test, and *p* < 0.05 is considered statistically significant.

## 5. Conclusions

We employed a TGF-β1-mediated cell fibrosis model for orbital fibroblasts and demonstrated that curcumin, at 1–5 ng/mL, was able to suppress TGF-β1-induced myofibroblast differentiation and to attenuate the pro-angiogenic activities of the primary culture of orbital fibroblasts. The experimental results have allowed us to suggest that curcumin may have a therapeutic effect in the treatment of GO.

## Figures and Tables

**Figure 1 ijms-22-06829-f001:**
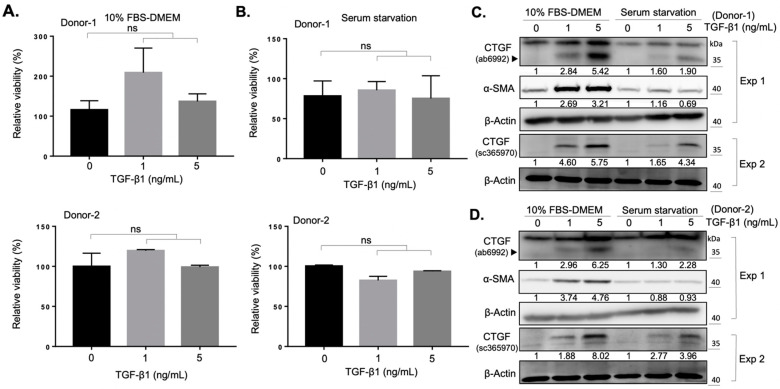
TGF-β1 promotes the myofibroblast differentiation of orbital fibroblasts. (**A**) The viability of fibroblasts from a normal subject (Donor-1) and a GO patient (Donor-2) was measured after treatment with the indicated doses of TGF-β1. Cells were seeded in 10% FBS-containing medium overnight and treated with TGF-β1 for an additional 24 h. Data are expressed as the mean ± SD (*n* = 3). ns, not significant. (**B**) Viability of the fibroblasts was measured after treatment with indicated doses of TGF-β1. Cells were seeded in the basal DMEM medium overnight and treated with TGF-β1 for an additional 24 h. Data are expressed as the mean ± SD (*n* = 3). ns, not significant. (**C**) Western blots showed the expression of CTGF and α-SMA at indicated culture conditions of the orbital fibroblasts from Donor-1 for 24 h. The relative intensity value to corresponding controls was indicated below. Exp 1: experiment 1; Exp 2: experiment 2. (**D**) Western blots showed the expression of CTGF and α-SMA at indicated culture conditions of the orbital fibroblasts from Donor-2 for 24 h. The relative intensity value to corresponding controls was indicated below. Exp 1: experiment 1; Exp 2: experiment 2.

**Figure 2 ijms-22-06829-f002:**
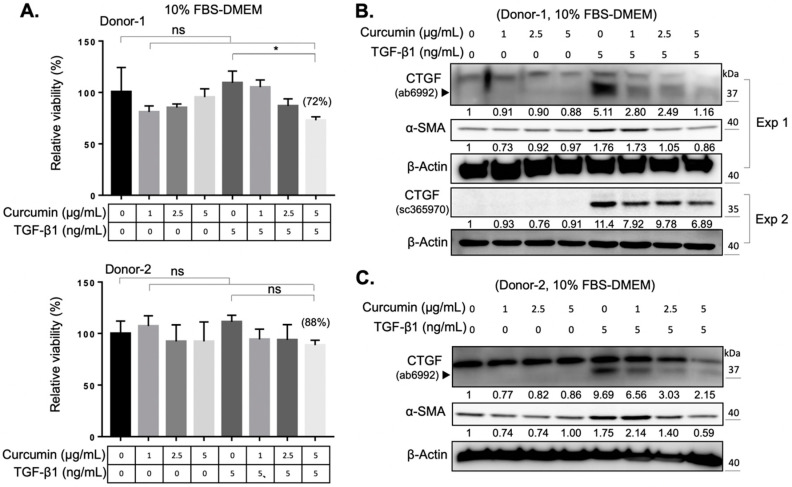
Curcumin suppresses the TGF-β1-activated myofibroblast differentiation. (**A**) The viability of orbital fibroblasts was measured after treatment with the indicated doses of curcumin and TGF-β1. Cells were seeded in 10% FBS-containing medium overnight and treated with the indicated dose of curcumin, TGF-β1 or both for an additional 24 h. The relative viability of orbital fibroblasts after treatment with 5 μg/mL of curcumin and 5 ng/mL of TGF-β1 is indicated in brackets. Data are expressed as the mean ± SD (*n* = 3). ns, not significant. *, *p* < 0.05. (**B**,**C**) Western blots showed the expression levels of CTGF and α-SMA at indicated culture conditions of the orbital fibroblasts from Donor-1 (**B**) and Donor-2 (**C**). The relative intensity value to corresponding controls is indicated below. Exp 1: experiment 1; Exp 2: experiment 2.

**Figure 3 ijms-22-06829-f003:**
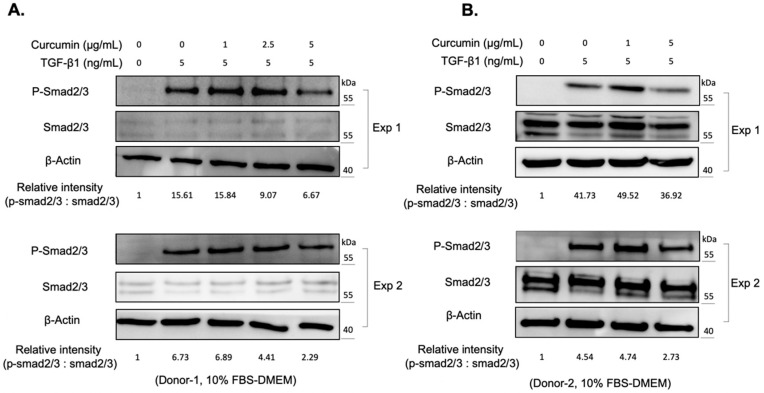
Curcumin attenuates the TGF-β1 signaling in orbital fibroblasts. (**A**) Western blots showed the expression of phosphorylated Smad 2/3 (p-Smad 2/3) and total Smad proteins (Smad 2/3) at indicated conditions of the orbital fibroblasts from Donor-1. The relative intensity of phosphorylated Smad 2/3 quantified using the Image J software. Exp 1: experiment 1; Exp 2: experiment 2. (**B**) Western blots showed the expression levels of phosphorylated Smad 2/3 (p-Smad 2/3) and total Smad proteins (Smad 2/3) at indicated conditions of the orbital fibroblasts from Donor-2. The relative intensity of phosphorylated Smad 2/3 quantified using the Image J software. Exp 1: experiment 1; Exp 2: experiment 2.

**Figure 4 ijms-22-06829-f004:**
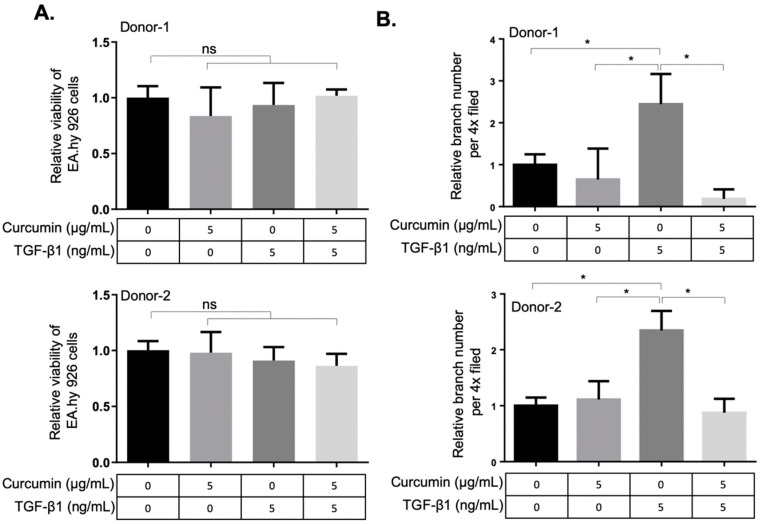
Curcumin suppresses the TGF-β1-mediated angiogenic activity in EA.hy926 endothelial cells. (**A**) Viability of EA.hy926 endothelial cells (ECs) was measured after treatment with conditioned medium from curcumin or TGF-β1-treated orbital fibroblasts. The fibroblasts were seeded in 10% FBS-containing medium overnight and then treated with curcumin or TGF-β1 for 24 h. The medium was then discarded and replenished with the basal DMEM for an additional 24 h as the conditioned medium for treatment of ECs. Data are expressed as the mean ± SD (*n* = 3). ns, not significant. (**B**) The relative tube-branching numbers of ECs cultured at indicated conditioned medium from orbital fibroblasts. (4× field) Data are expressed as the mean ± SD (*n* > 3). *, *p* < 0.05.

## Data Availability

The data supporting the findings of this study are available in the Appendix A online.

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
