# Peer review of "Curcumin Suppresses TGF-β1-Induced Myofibroblast Differentiation and Attenuates Angiogenic Activity of Orbital Fibroblasts"

_ijms, 2021, doi:10.3390/ijms22136829_

Round 1
Reviewer 1 Report
Dear authors,
The manuscript present a relevant results. However, the structure of manuscript is not correct. The authors have included information in the results section that are not results, but rather discussion of the work. In addition, some sections of the material and methods section are not well described. Finally, the authores have carried out parametric tests without previously checking if the data were distributed in a normal way and presented homoscedasticity. For these reasons, I recommend rejecting the paper for publication.
Reviewer 2 Report
Please see the attachment and below:
- Use a space before the references in the text.
- In case of all western blot data figures, please mention the band size.
- Figure 1C-D (figure legends), please mention the time points (how long the cells were treated with samples).
- Please include statistical analysis for all western blot data.
- Figure 3A, why the total smad proteins (smad 2/3) bands are not clear whereas, Figure 3B is clear?
- Mark the TGF-β1-enhanced tube-forming capacity, transwell migration capacity, tube branching numbers in the images.
- Line 159, Data are expressed as mean ± SD (n=2). Why n=2 only? That means this data is not checked for statistical significance. But why?
- Line 216, β-actin (#A1978) were purchased from Sigma-Aldrich Chemical Co. Antibody? Or protein?
- Line 261, the RIPA lysis buffer. Mention the recipe.
- Line 271-272, The protein signals were visualized by enhanced chemiluminescence (ECL). Mention some more details i.e., film or machines used etc.
- Most of the experiments were done with n=3 and nothing mentioned about the repeat of the experiments. This is improper for making conclusion. However, the authors have rights to answer. So, I am recommending ‘Reconsider after major revision’.

Reviewer 3 Report
In this manuscript, Yu and colleagues have studied the role of curcumin in suppressing myofibroblast differentiation of orbital fibroblasts. They have performed in vitro assays on fibroblasts isolated from human donors and found that curcumin suppresses TGFβ signalling. They have additionally shown that curcumin treated fibroblasts elicit a reduced angiogenic response.
The authors have performed experiments with attention to technical details, but the following points need to be addressed to strengthen their conclusions.
- The authors have performed near identical experiments from fibroblasts derived from normal and those from a patient with GO. GO samples, as the authors have mentioned, are expected to have a higher composition of myofibroblasts, and are not supposed to be grouped together with normal samples. The authors need to explain the rationale behind their experimental design. Ideally, they need to repeat their experiments with fibroblasts derived an additional healthy donor.
- The authors have shown that curcumin suppresses TGFβ1 signalling by looking at CTGF levels. They need to further validate this observation by looking at additional TGFβ1 downstream factors.
- Quality of bright-field images shown in Fig. S1 must be improved. Many regions appear to be out of focus and shadows are seen.
- The choice of EA.hy926 endothelial cells must be explained. The authors need to validate their results on an additional endothelial cell line.
- It would be interesting to know if curcumin treatment reverses the myofibroblast conversion of orbital fibroblasts.
Round 2
Reviewer 1 Report
No further comments
Author Response
"Please see the attachment."

Reviewer 2 Report
Please see the attachment.

Author Response
"Please see the attachment."

Reviewer 3 Report
They authors have not well addressed point #1 of my previous comments. The article in the present form has just one normal sample and one GO sample included in the analysis. It is necessary to have a minimum of n=2 samples/condition.
Author Response
"Please see the attachment."

Round 3
Reviewer 3 Report
Accept in present form.